# Hospitalisations Related to the Combination of ACE Inhibitors and/or Angiotensin Receptor Blockers with Diuretics and NSAIDs: A Post Hoc Analysis on the Risks Associated with Triple Whammy

**DOI:** 10.3390/healthcare11020238

**Published:** 2023-01-12

**Authors:** Irene Mattioli, Alessandra Bettiol, Giada Crescioli, Roberto Bonaiuti, Guido Mannaioni, Alfredo Vannacci, Niccolò Lombardi

**Affiliations:** 1Department of Experimental and Clinical Medicine, University of Florence, 50139 Florence, Italy; 2Department of Neurosciences, Psychology, Drug Research and Child Health, Section of Pharmacology and Toxicology, University of Florence, 50139 Florence, Italy; 3Tuscan Regional Centre of Pharmacovigilance, 50139 Florence, Italy; 4Toxicology Unit, Poison Control Center, Careggi University Hospital, 50139 Florence, Italy

**Keywords:** ACE inhibitors, angiotensin-II-receptor antagonists, diuretics, nonsteroidal anti-inflammatory drugs, renal kidney injury, triple whammy, hospitalisations, adverse drug reactions, drug safety, pharmacovigilance

## Abstract

This post hoc analysis aimed to assess and characterise adverse events (AEs) related to the triple whammy (i.e., combination therapy of ACE inhibitors, ACE-I, and/or angiotensin receptor blockers, ARBs, with diuretics and non-steroidal anti-inflammatory drugs, NSAIDs) leading to emergency department (ED) visits and/or hospitalisations in the Italian setting. The MEREAFaPS database was analysed. ED visits related to co-treatment with ACE-I and/or ARBs, diuretics, and NSAIDs were considered. Information on the AE (including classification, seriousness, and outcome), suspected and concomitant drugs, and concomitant conditions was retrieved and analysed. Logistic regression was used to estimate the reporting odds ratios (RORs) of hospitalisation associated with the drugs of interest. Between 1 January 2007, and 31 December 2018, 80 patients visited the ED for AEs related to the triple whammy, and a total of 261 suspected drugs were involved. Patients were mostly Caucasian females, with a median age of 85 years, and only 9 of them had renal manifestations. In this subset, drug–drug interaction contributed to kidney injury. Most patients presented a Charlson comorbidity index of 4–5. Overall, 47 patients were hospitalised (58.75%), but no significant differences in the risk of hospitalisation were found according to demographic, clinical, or therapeutic features.

## 1. Introduction

Polypharmacy, i.e., the concomitant daily assumption of multiple medications (≥5), is generally a consequence of chronic co-existing diseases, each of which requires drug treatment. This condition is most common among the elderly, affecting about 50% of the population aged over 65 [1]. It is well known that, especially in frail subgroups, polypharmacy can lead to serious adverse events (AEs), poor adherence to pharmacological and non-pharmacological therapy, and numerous drug–drug and drug–disease interactions [2,3].

In this regard, the elderly are at a higher risk of incurring AEs due to drug interactions not only because of the number of drugs taken but also because of the natural ageing process. In fact, increasing age is associated with physiological changes such as decreased kidney and liver function, which can negatively influence drug pharmacodynamics and pharmacokinetics [4]. Among interactions that can result in serious AEs, the literature describes the “triple whammy”, a term coined in 2000 by Australian clinicians and researchers to define the co-treatment with ACE inhibitors (ACE-I) and/or angiotensin receptor blockers (ARBs), in combination with diuretics and non-steroidal anti-inflammatory drugs (NSAIDs) or cox-2 inhibitors [5]. Subsequently, studies have further explored the risk of AEs resulting from this co-treatment, suggesting an increased risk of acute kidney injury (AKI) in patients on the triple whammy [6,7].

This is probably due to the inhibition of the renin–angiotensin system by ACE-I or ARBs that dilate efferent arterioles, leading to a decrease in the glomerular filtration rate (GFR) and renal function. Moreover, co-treatment with diuretics can lead to hypovolemia, thus reducing renal plasma flow and increasing the risk of renal failure [8]. The risk of AKI is further increased when this polypharmacy is combined with NSAIDs: indeed, NSAIDs are known to block the cyclooxygenase involved in the synthesis of prostacyclins, prostanoids, and prostaglandins, which plays a key role in the physiological regulation of several renal processes, such as ions and water transport through the nephron, and the dilation of the afferent arteriole by increasing GFR [9].

The present analysis aimed to assess and characterise AEs associated with the triple whammy as a cause of emergency department (ED) visits and/or hospitalisations in the Italian setting, focusing on the risk of AKI related to the concomitant use of ACE-I and/or ARBs, in combination with diuretics and NSAIDs.

## 2. Materials and Methods

### 2.1. Study Design

An observational retrospective post hoc analysis was conducted on the MEREAFaPS database, an Italian database of pharmacovigilance report forms of suspected AEs requiring ED visits, actively collected in ED between 1 January 2007, and 31 December 2018.

### 2.2. Study Setting

The present analysis included 80 patients (out of 61,855 cases collected in the MEREAFaPS database) exposed to the triple whammy who visited the ED of 94 general hospitals distributed in five Italian Regions (Lombardy, Piedmont, Tuscany, Emilia-Romagna, and Campania). More details, including the methodology applied for the construction of the MEREAFaPS database, can be found in our previous publications [10,11,12,13,14,15,16]. As already described, all pharmacovigilance report forms analysed here were all validated in terms of causality assessment using specific algorithms and/or scales (i.e., Naranjo scale) as requested by the Italian Medicines Agency and by the Italian Pharmacovigilance System legislation [10,11,12,13,14,15,16]. Furthermore, a multidisciplinary team composed by experts in clinical pharmacology and toxicology, and pharmacoepidemiology, performed a thorough evaluation of cases included in the MEREAFaPS database, in order to maintain a high-quality standard of the data.

### 2.3. Suspected Medications

Suspected medications were classified according to the Anatomical Therapeutic Chemical (ATC) classification system. Only pharmacovigilance report forms with ACE-I (ATC classes: C09A*/C09B*) and/or ARBs (ATC classes: C09C*/C09D*), diuretics (ATC classes: C03*) and NSAIDs (including cox-2 inhibitors) (ATC classes: M01A*) as suspected agents were considered. Patients who developed an AE for any other reason than the exposure to ACE-I and/or ARB, diuretics, or NSAIDs (including cox-2 inhibitors) were excluded. Suspected drugs were described by pharmacological classes and active principles. Information on demographic and clinical features and on concomitant drugs and conditions was also retrieved.

The patient’s clinical complexity was estimated by calculating the Charlson comorbidity index [17], a weighted index to predict risk of death within 1 year of hospitalisation for patients with specific comorbidities. Nineteen conditions were included in the index (i.e., myocardial infarction, congestive heart failure, peripheral vascular disease, cerebrovascular accident or transient ischemic attack, dementia, chronic obstructive pulmonary disease, connective tissue disease, peptic ulcer disease, liver disease, diabetes mellitus, hemiplegia, moderate to severe chronic kidney disease, solid tumour, leukaemia, lymphoma, AIDS, and COVID-19). Each condition was assigned a weight from 1 to 6, based on the estimated 1-year mortality hazard ratio from a Cox proportional hazards model. These weights were summed to produce the Charlson comorbidity score. During the last decades, some researchers adapted the Charlson index to ICD-9-CM diagnosis and procedure codes so that the index could be calculated using administrative data.

### 2.4. Adverse Events

AEs were coded using the Medical Dictionary for Regulatory Activities (MedDRA) and Preferred Term (PT) [18]. AEs requiring patient’s hospitalisation were considered as serious. AE outcomes were assessed and classified as resolution with sequelae, still unresolved, complete resolution, improvement, death, and not available. When available in the pharmacovigilance report form, information on drug misuse, abuse, medication error or overdose, defined according to the Good Pharmacovigilance Practices (GVP)—Module VI (Rev 2), was also collected [19].

### 2.5. Drug–Drug Interactions

Each case was evaluated with the aim of identifying the presence of drug–drug interactions, which may have contributed to AEs. Drug–drug interactions were identified using two different validated tools: (1) the open access Drug Interaction Checker [20], and (2) the drug interaction software IBM Micromedex^®^ (Thomson Reuters Healthcare Inc., Greenwood Village, CO, USA), available online with restricted access [21]. As reported in the IBM Micromedex^®^ and Drug Interaction Checker tools, drug–drug interactions were classified as mild, moderate, or major, depending on their clinical impact on patient.

### 2.6. Statistical Analysis

Descriptive statistics were used to summarise data. Categorical data were reported as frequencies and percentages, whereas continuous data were reported as median values with the related interquartile ranges (IQRs).

Within the triple whammy group, multivariate logistic regression models were fitted to estimate the reporting odds ratios (RORs) with 95% confidence intervals (CIs) of hospitalisation according to age, sex, concomitant conditions, and concomitant medications. The presence of multicollinearity in the multivariate regression model was excluded by calculating the variance inflation factor (VIF). All results were considered statistically significant at *p* < 0.05.

Data management and statistical analysis were performed with STATA v17 software (StataCorp, USA).

## 3. Results

In the period between 1 January 2007, and 31 December 2018, a total of 80 pharmacovigilance report forms resulting in ED visits and related to the triple whammy were collected.

Table 1 describes demographic characteristics of these cases. Of the 80 patients, 51 were female (63.7%) and 78 were Caucasians (97.5%), with a median age of 85 years (IQR 25-98). In most cases, three suspected agents were involved, mostly administered orally. The presence of four or five (i.e., polypharmacy) suspected agents was reported in 26.3%. No case of drug abuse, misuse, overdose, or medication error was reported. Forty-seven AEs were classified as serious (58.8%), as they required patient hospitalisation. However, AEs generally resulted in an improvement (50.0%) or a complete resolution of symptoms (40.0%).

Table 2 reports the suspected drugs classes (n = 261) involved in these 80 pharmacovigilance report forms. The most frequently reported suspected drug classes were diuretics (35.6%) followed by NSAIDs (32.9%), ACE-I (21.5%) and ARBs (9.9%). Out of 261 suspected agents, 156 were involved in the 47 reports of serious AEs.

Information on concomitant (not suspected) medications and concomitant conditions is available in Table 3. More than 87% of the patients were concomitantly treated with additional drugs other than those included in the definition of the triple whammy. The most frequently reported concomitant medications included cardiovascular drugs (53.7%) and anticoagulant/antiplatelet medications (37.5%), followed by antipsychotics or antidepressants (31.2%), statins (27.5%), and antidiabetic drugs (26.2%). Drugs reported to a lesser extent were gout medications (16.2%), drugs affecting the thyroid (8.7%), medications for respiratory disorders (8.7%), and prostate drugs (5%).

Coherently, when considering concomitant conditions, diabetes was the most frequently reported one (26.2%), followed by cardiac and respiratory disorders (13.7% and 7.5%, respectively), and rheumatologic, thyroid, coagulation, and psychiatric disorders (5%).

Most patients included in this analysis presented a Charlson comorbidity index of 4 and 5 (25% and 47.5%, respectively).

Table 4 shows the clinical manifestations presented by the 80 patients with AEs related to the triple whammy. A total of 268 different manifestations were reported, of which 160 occurred in patients requiring hospitalisations. The most frequent AEs were related to gastrointestinal symptoms (12.7%), skin disorders (9.3%), neurological symptoms (5.5%), and metabolic and nutrition disorders (4.1%).

Table 5 describes the demographic characteristics of the nine cases with renal manifestations. Only 9/80 patients (11.3%) had a renal manifestation, including 5 females (55.6%), all Caucasian patients, with a median age of 84 years (IQR 61-106). In all cases, three suspected agents were involved, mostly administered orally (88.9%). All AEs were related to drug–drug interaction, and no case of drug abuse, misuse, overdose, or medication error was reported. Seven AEs were classified as serious (77.8%), as they required patient hospitalisation. Fortunately, AE generally resulted in a complete resolution of symptoms or improvement (55.6% and 44.4% respectively).

Appendix A summarises the estimated risks of hospitalisation for all patients on the triple whammy, according to age, sex, concomitant conditions, and concomitant medications. However, the calculated estimates, probably due to the small sample of cases analysed, did not show any increased risk of hospitalisation associated with the concomitant use of the drugs of interest. Nevertheless, we believe it might be useful for the reader to provide a trend on the association between the considered variables and the outcome.

## 4. Discussion

This post hoc analysis, conducted on a large Italian active pharmacovigilance database, characterised AEs related to the exposure to the triple whammy leading to ED visit and/or hospitalisations, with a focus on the AEs related to AKI.

Of notice, on a total of more than 60,000 pharmacovigilance report forms included in the MEREAFaPS database, only 80 patients reported AEs potentially related to the triple whammy. In our opinion, this evidence could be considered a first interesting result. If at first glance 80 patients may seem very few for such a long period of study (12 years), questioning the validity of the original database considering the little evidence published in the scientific literature on the triple whammy at the ED level, our results can highlight two important aspects: (1) very little is still known about this medical occurrence (triple whammy) associated with polypharmacy; (2) in our country, Italy, polytherapy based on diuretics, ACE-I and/or ARBs, and NSAIDs is quite appropriate and well controlled, especially in adults aged >80 years. Indeed, most of the patients analysed here were elderly (61.3% being aged 80 or more) and of female sex (63.7%). It is well known that old age (+80 years) is a major risk factor for renal failure, particularly when cardiovascular disease and diabetes are present [9]. For this reason, it is important that pharmacological therapies are administered appropriately, especially in clinically complex subjects.

Notably, cardiovascular disease and diabetes were the most frequently reported concomitant diseases in our cohort. Nevertheless, only 9 out of 80 patients reported AKI (11%). This result agrees with a recently published Japanese study on spontaneous reports of adverse reactions, highlighting that only about 5% of the patients accessing the ED for triple whammy-related AEs had renal manifestations [9,22]. Noteworthy, in all patients with renal manifestations, drug–drug interactions contributed to the event. Moreover, in our cohort, most of the triple whammy-related AEs leading to ED visit were associated with drug–drug interactions (86.3%, the majority of which defined as moderate), and the most common manifestations were acute renal failure and/or electrolyte abnormalities (i.e., hyponatremia, hyperkalaemia) in 50% and 25% of hospitalised patients, respectively.

When focusing on the most involved drugs in our cohort, a total of 261 drugs implicated in the triple whammy were reported. As already described in a New Zealand study and in one recently published by Piekarska M et al., furosemide was the most frequently reported diuretic agent (16.1%), enalapril and ramipril the most reported ACE-I (8.43% and 6.9%, respectively), while among the most implicated NSAIDs were ibuprofen and diclofenac (8% and 6.9%, respectively) [23,24].

Overall, our data suggest that prescribing NSAIDs to patients already receiving ACE-I or ARBs with a diuretic should be well controlled and, when possible, avoided. In addition, these results underscore the importance of educating both healthcare professionals (i.e., general practitioners, community pharmacists, etc.) and patients, making sure they know to avoid over-the-counter medications, particularly NSAIDs.

Dietary supplements containing antioxidants (i.e., vitamin C) have been shown to be reno- and cardio-protective agents and, for this reason, they can be considered a valid solution to decrease the incidence of renal damage in patients who consume diuretics, ACE-I and/or ARBs and NSAIDs either separately or in combination. Oral administration of rutin and vitamin C showed a favourable synergistic effect and could, in the future, prove effective in reducing inflammation and oxidative stress in end-stage renal patients [25]. Moreover, supplementation with omega-3 in patients with end-stage renal disease showed a highly significant reduction in total cholesterol and a highly significant increase in glutathione peroxidase and superoxide dismutase levels proving the omega-3′ beneficial effect on serum lipid profile and oxidative stress [26]. Silymarin with its antioxidant and anti-inflammatory properties was used as an adjuvant therapy for glycaemic control, lipid profile, and insulin resistance [27] Additionally, the micronised purified flavonoid fraction of *Rutaceae aurantiae*, at the daily dose of 500 mg, proved to be helpful in reducing glucose level and the risk of cardiovascular disease [28]. The co-administration of these antioxidant products could be recommended in the real-world practice to reduce the clinical burden of such cardiovascular and anti-inflammatory pharmacological therapies.

Considering that our results were collected through a national project of active pharmacovigilance, these data suggest that an active approach can be considered a valuable option to better define the safety profile of commercial drugs, especially those associated with a higher risk profile in frail populations (i.e., elderly). This consideration is even more relevant when considering both the appropriate prescription and appropriate use of medications in the general population.

In conclusion, according to our analysis, in the presence of a therapy based on diuretics, ACE-I and/or ARBs, and NSAIDs, greater awareness and attention should be paid both by prescribers and patients, especially those with one or more comorbidities.

### Limitations and Strengths

The present analysis has some limitations. First, an underestimation of the AE rate cannot be completely excluded, as not all patients who present an AE go to the ED or report it spontaneously. In any case, considering that data on pharmacovigilance report forms were collected through an active pharmacovigilance project, the issue of underreporting can be considered of relatively low significance. Moreover, there are other important aspects to report. First, some of the comorbidities were traced from the patients’ drug therapy because such data were not always present in the pharmacovigilance report forms. Second, it was not possible to determine whether the occasional, short-term use of an NSAID in a patient receiving an ACE- I and/or an ARB with a diuretic carries an increased risk of renal damage. Third, most patients exposed to the triple whammy suffered from numerous concomitant diseases and co-treatments, making it quite difficult to attribute the reported AEs (especially those related to renal manifestations) solely to the interactions between the drugs objective of the analysis. Noteworthy also is the use of dietary supplements and integrative medicine products which, by interacting with other prescription drugs and with the underlying pathologies, may have contributed to the events of interest. Furthermore, the relatively small sample size of the patients evaluated in this analysis could not be considered representative of the real burden of the triple whammy in the Italian general population, particularly at the ED level. As a final point, the results of the multivariate analysis are limited by the small number of events reported in this population.

Despite these limitations, this is the first study investigating the risk of renal injury associated with the triple whammy and conducted in Italy on AEs associated with ED visits and hospitalisations. In addition, our study involved several EDs distributed through the territory, so the evidence reported here can be considered an interesting explorative analysis of this clinical occurrence as cause of ED visits and/or hospitalisation.

## 5. Conclusions

To date, few studies are available investigating the risk of renal injury associated with the triple whammy. Further investigations need to be made both to assess the safety profile of diuretics, ACE-I and/or ARBs, and NSAIDs in combination, and to identify even more definitively patients who should not be subjected to such polypharmacy because of increased renal risk. Moreover, these perspectives should be made to involve a more representative sample of patients. Noteworthy, polypharmacy is not necessarily a problem, especially in the elderly. Of course, inappropriate polypharmacy should be avoided in frail subgroups. Current international guidelines recommend four drugs as basic therapy for congestive heart failure [29]; similar recommendations have been applied to type 2 diabetes [30] and chronic coronary syndrome [31]. These individuals will invariably have (and should have) polypharmacy. The lack of appropriate and controlled polypharmacy would indicate a sub-therapeutic management of these clinically relevant diseases.

The importance of active pharmacovigilance over spontaneous reporting of AEs may be one of the main keys to improve the general knowledge of the safety profile of medications associated with the triple whammy in clinical practice, and the ED can be considered a privileged observatory for this purpose.

## Figures and Tables

**Table 1 healthcare-11-00238-t001:** Study population.

Case Characteristics	No. of Cases 80 (%)	No. of Cases Resulting in Hospitalisation 47 (%)
**Patient age, years**		
<80	31 (38.8)	17 (36.2)
≥80	49 (61.3)	30 (63.8)
**Sex**		
Female	51 (63.7)	27 (33.7)
Male	29 (36.2)	20 (25.0)
**Patient ethnicity**		
Caucasian	78 (97.5)	45 (95.7)
Other	2 (2.5)	2 (4.2)
**No. of suspected drugs involved in the AE**		
3	57 (71.3)	34 (72.3)
4–5	21 (26.3)	11 (23.4)
>5	2 (2.5)	2 (4.2)
**Administration route**		
Enteral (oral)	64 (80.0)	34 (72.3)
Parenteral	16 (20.0)	13 (27.7)
**Adverse events due to**		
Abuse/misuse	-	-
Drug–drug interaction *	69 (86.3)	39 (78.7)
Overdose	-	-
Therapeutic error	1 (0.8)	-
**Outcome**		
Complete resolution	32 (40.0)	20 (42.5)
Improved	40 (50.0)	21 (44.7)
Still unresolved	-	-
Resolution with sequelae	-	-
Death	-	-
Not available	8 (10.0)	6 (12.8)

* Drug–drug interactions contributed to 29 events (42.0% out of 69), of which 24 (51.1% out of 39) resulted in hospitalisation. The majority of interactions were reported in the pharmacovigilance report forms at the time of the AE occurrence. Applying specific tools, most of interactions were defined as “moderate”.

**Table 2 healthcare-11-00238-t002:** Suspected pharmacological classes.

Drug Class	ED Visits for AEs	ED Visits for AEs Resulting in Hospitalisation
	No. of Suspected Agents 261 (%)	No. of Suspected Agents 156 (%)
**Diuretics (n = 93)**		
Furosemide	42 (16.1)	25 (16.0)
Torasemide	11 (4.2)	5 (3.2)
Amiloride hydrochloride + Hydrochlorothiazide	11 (4.2)	5 (3.2)
Potassium canrenoate	8 (3.1)	4 (2.6)
Furosemide + Spironolactone	7 (2.7)	4 (2.6)
Spironolactone	6 (2.3)	6 (3.8)
Chlorthalidone	5 (1.9)	4 (2.6)
Hydrochlorothiazide	3 (1.2)	2 (1.8)
**NSAIDs (n = 86)**		
Ibuprofen	21 (8.0)	13 (8.33)
Diclofenac	18 (6.9)	13 (8.33)
Ketoprofen	9 (3.5)	7 (4.5)
Ketorolac	8 (3.1)	6 (3.9)
Etoricoxib	6 (2.3)	2 (1.3)
Indomethacin	5 (1.9)	1 (0.6)
Celecoxib	4 (1.5)	3 (1.3)
Nimesulide	4 (1.5)	2 (1.3)
Others	11 (4.2)	5 (3.2)
**ACE inhibitors (n = 56)**		
Enalapril	22 (8.4)	12 (7.7)
Ramipril	18 (6.9)	16 (10.3)
Perindopril	6 (2.3)	2 (1.3)
Others	10 (3.8)	5 (3.2)
**ARBs (n = 26)**		
Valsartan	6 (2.3)	1 (0.6)
Losartan	5 (1.9)	3 (1.9)
Telmisartan	5 (1.9)	5 (3.2)
Candesartan cilexetil	3 (1.2)	1 (0.6)
Others	7 (2.7)	4 (2.6)

ACE: angiotensin converting enzyme; ARB: angiotensin receptor blocker; NSAID: non-steroidal anti-inflammatory drug.

**Table 3 healthcare-11-00238-t003:** Concomitant medications and conditions.

	No. of Case 80 (%)	No. of Cases Resulting in Hospitalisation 47 (%)
**Concomitant medications**		
No	10 (12.5)	7 (14.9)
Yes	70 (87.5)	40 (85.1)
**Most frequently reported medications**		
Cardiac therapy	43 (53.8)	24 (51.1)
Blood therapy	30 (37.5)	20 (42.6)
Antipsychotics/antidepressants	25 (31.3)	11 (23.4)
Statins	22 (27.5)	15 (31.9)
Antidiabetic therapy	21 (26.3)	9 (19.2)
Gout therapy	13 (16.3)	7 (14.9)
Thyroid therapy	7 (8.8)	5 (10.6)
Respiratory therapy	7 (8.8)	5 (10.6)
Prostate therapy	4 (5.0)	3 (6.4)
**Concomitant conditions**		
No	26 (32.5)	17 (21.3)
Yes	54 (67.5)	30 (37.5)
**Most frequently reported conditions**		
Diabetes	21 (26.3)	9 (11.3)
Cardiac disorders	11 (13.8)	8 (8.8)
Respiratory disorders	6 (7.5)	2 (2.5)
Rheumatologic disorders	4 (5.0)	2 (2.5)
Thyroid disorders	4 (5.0)	3 (3.8)
Clotting disorders	4 (5.0)	3 (3.8)
Anxiety and depressive disorders	4 (5.0)	1 (1.3)
**Charlson comorbidity index**		
1	3 (3.8)	-
2	1 (1.3)	-
3	5 (6.3)	-
4	20 (25.0)	2 (4.3)
5	38 (47.5)	3 (6.4)
6	10 (12.5)	3 (6.4)
7	3 (3.8)	1 (2.1)

**Table 4 healthcare-11-00238-t004:** Manifestations of adverse events.

Adverse Event Manifestations (MedDRA SOC Term)	No. of AEs Leading the Patient in the ED 268 (%)	No of AEs in ED Resulting in Hospitalisation 160 (%)
**Metabolism and nutrition disorders**		
Hyperkalaemia	5 (1.9)	5 (3.1)
Hyponatremia	4 (1.5)	4 (1.5)
Others	2 (0.8)	1 (0.6)
**Immune system disorders**		
Allergic reaction	5 (1.9)	3 (1.9)
Others	1 (0.4)	0 (0)
**Hematopoietic system disorders**		
Anaemia	6 (2.2)	6 (3.8)
Others	4 (1.5)	1 (0.6)
**Nervous system disorders**		
Lipothymia	2 (0.8)	0 (0)
Syncope	2 (0.8)	2 (1.3)
Dizziness	2 (0.8)	1 (0.6)
Loss of consciousness	2 (0.8)	2 (1.3)
Light-headedness	7 (2.6)	4 (1.5)
**Skin and subcutaneous tissue disorders**		
Generalised itching	7 (2.6)	2 (1.3)
Urticarial rash	7 (2.6)	2 (1.3)
Erythema	4 (1.5)	1 (0.6)
Others	7 (2.6)	2 (1.3)
**Gastrointestinal disorders**		
Melena	4 (1.5)	4 (1.5)
Epigastralgia	3 (1.1)	2 (1.3)
Diarrhoea	3 (1.1)	3 (1.9)
Gastric ulcer with haemorrhage	3 (1.1)	2 (1.3)
Vomiting	3 (1.1)	1 (0.63)
Others	18 (6.7)	15 (9.38)
**Kidney and urinary diseases**		
Acute over chronic renal failure	5 (1.9)	5 (3.1)
Acute renal failure	2 (0.75)	1 (0.6)
Impaired renal function	1 (0.4)	0 (0)
Anuria	1 (0.4)	1 (0.6)
**Systemic diseases and conditions related to the site of administration**		
Facial oedema	3 (1.12)	0 (0)
Drug interaction	3 (1.12)	0 (0)
Others	8 (3.0)	2 (1.3)
**Vascular disorders**	3 (1.1)	1 (0.6)
**Trauma, poisoning, and procedure complications**	3 (1.1)	3 (1.9)
**Respiratory, thoracic, and mediastinal disorders**	2 (0.8)	2 (1.25)
**Hepatobiliary disorders**	1 (0.4)	0 (0)
**Eye diseases**	1 (0.4)	1 (0.63)
**Musculoskeletal and tissue disorders**	1 (0.4)	0 (0)
**Psychiatric disorders**	5 (1.9)	4 (1.5)
**Diagnostic tests**	3 (1.1)	2 (1.3)
**Cardiac disorders**	4 (1.5)	2 (1.3)

**Table 5 healthcare-11-00238-t005:** Demographic characteristics of the nine cases with renal manifestations.

Case Characteristics	No. of Case 9 (%)	No. of Cases Resulting in Hospitalisation 7 (%)
**Patient Age, Years**		
<80	3 (33.3)	2 (28.6)
≥80	6 (66.7)	5 (71.4)
**Sex**		
Female	5 (55.6)	1 (14.3)
Male	4 (44.4)	1 (14.3)
**Patient ethnicity**		
Caucasian	9 (100)	7 (100)
Other	-	-
**No. of suspected drugs involved in the AE**		
<4	9 (100)	7 (100)
≥4	-	-
**Administration route**		
Systemic	1 (11.1)	1 (14.3)
Oral	8 (88.9)	2 (28.6)
**Adverse events due to**		
Abuse/misuse	-	-
Drug–drug interaction *	9 (100)	7 (100)
Overdose	-	-
Therapeutic error	-	-
**Outcome**		
Complete resolution	5 (55.6)	1 (14.3)
Improved	4 (44.4)	1 (14.3)
Still unresolved	-	-
Resolution with sequelae	-	-
Death	-	-

* Drug–drug interactions contributed to eight events (88.9% out of nine), of which seven (100% out of seven) resulted in hospitalisation. The majority of interactions were reported in the pharmacovigilance report forms at the time of the AE occurrence. Applying specific tools, most of interactions were defined as “moderate”.

## Data Availability

Data that support the findings of this study are available upon reasonable request from the corresponding author, G.C.

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
