# Peer review of "Hospitalisations Related to the Combination of ACE Inhibitors and/or Angiotensin Receptor Blockers with Diuretics and NSAIDs: A Post Hoc Analysis on the Risks Associated with Triple Whammy"

_healthcare, 2023, doi:10.3390/healthcare11020238_

Round 1

Reviewer 1 Report

The manuscript entitled ‘Hospitalisations related to the combination of ACE inhibitors and/or sartans with diuretics and NSAIDs in Italy: a statement on the risks of kidney injury associated with the so-called “triple whammy’ considers the possible risks (as hospital and ED visits) associated with a polypharmacy approach to patient management, particularly for patients with AKI. Although this is a descriptive study without a working hypothesis to test, there is merit in a study covering the potential adverse effects of polypharmacy, typically observed in the elderly.   The manuscript provides a series of tables that are suitably explained and culminates with a reporting odds ratio figure, that provides details of trends, and highlights the variability within the cohort.  There are some minor points for the authors to address in a revised manuscript:

Table 2, some of the drug names are written in Italian and not English equivalents, could this be changed where appropriate?

It would be useful to provide the reader with full details of the Charlson Comorbidity index.

Check through for typographical errors, such as Table 4, lightheadedness

The Figure 1 legend is repeated three times.

The authors consider the relatively small sample size and therefore limitation for mapping to the whole Italian cohort. However, in the methods section, the authors conflict this with the statement that “As already described [10-16], this data can be considered representative of the whole Italian general population”.  This conflict in written wording needs to be resolved.

Author Response

Reviewer #1:

The manuscript entitled ‘Hospitalisations related to the combination of ACE inhibitors and/or sartans with diuretics and NSAIDs in Italy: a statement on the risks of kidney injury associated with the so-called “triple whammy’ considers the possible risks (as hospital and ED visits) associated with a polypharmacy approach to patient management, particularly for patients with AKI. Although this is a descriptive study without a working hypothesis to test, there is merit in a study covering the potential adverse effects of polypharmacy, typically observed in the elderly. The manuscript provides a series of tables that are suitably explained and culminates with a reporting odds ratio figure, that provides details of trends, and highlights the variability within the cohort. 

We would like to thank Reviewer 1 for his/her consideration regarding our paper.

There are some minor points for the authors to address in a revised manuscript:

Table 2, some of the drug names are written in Italian and not English equivalents, could this be changed where appropriate?

Resolved.

It would be useful to provide the reader with full details of the Charlson Comorbidity index.

Resolved.

Check through for typographical errors, such as Table 4, lightheadedness

Resolved.

The Figure 1 legend is repeated three times.

Resolved.

The authors consider the relatively small sample size and therefore limitation for mapping to the whole Italian cohort. However, in the methods section, the authors conflict this with the statement that “As already described [10-16], this data can be considered representative of the whole Italian general population”.  This conflict in written wording needs to be resolved.

Resolved.

Reviewer 2 Report

more work to improve this paper 

Title: there is no scientific name “Sartans”, please change to Angiotensin Receptor Blockers throughout the manuscript 

 Indicate study type in the Title……: A retrospective cohort study 

 Abstract need to be structured: background, method, results, conclusion 

Introduction is week, revise as follows

 Polypharmacy, i.e., the concomitant daily assumption of multiple medications (5), 41 is generally a consequence of chronic co-existing diseases, each of which requires drug 42 treatment. This condition is most common among the elderly, affecting about 50% of the 43 population aged over-65 [1].

Polypharmacy can lead to serious adverse events (AEs), poor adherence to therapy, 45 and numerous drug-drug interactions [2-3].

These 2 paragraphs need to be combined into 1 paragraph

In this regard, the…..

Among these interactions

Combine these 2 paragraphs as well

There are so many tables, try to reduce them and move tables with unnecessary details to the supplementary 

Author Response

Reviewer #2:

More work to improve this paper

We would like to thank Reviewer 2 for his/her consideration regarding our paper.

Title: there is no scientific name “Sartans”, please change to Angiotensin Receptor Blockers throughout the manuscript

Resolved.

Indicate study type in the Title……: A retrospective cohort study

Resolved.

Abstract need to be structured: background, method, results, conclusion

Editorial guidelines require an unstructured abstract.

Introduction is week, revise as follows

Polypharmacy, i.e., the concomitant daily assumption of multiple medications (≥5), 41 is generally a consequence of chronic co-existing diseases, each of which requires drug 42 treatment. This condition is most common among the elderly, affecting about 50% of the 43 population aged over-65 [1].

Polypharmacy can lead to serious adverse events (AEs), poor adherence to therapy, 45 and numerous drug-drug interactions [2-3].

These 2 paragraphs need to be combined into 1 paragraph

Resolved.

In this regard, the…..

Among these interactions

Combine these 2 paragraphs as well

Resolved.

There are so many tables, try to reduce them and move tables with unnecessary details to the supplementary

Considering the small sample of patients analysed, we think that the number of tables is adequate for the description of their main clinical and pharmacological characteristics.

We moved Figure 1 in the Supplementary section.

Reviewer 3 Report

The manuscript is dealing with a very important topic by considering the adverse effects that might occur through polypharmacy and lead to kidney injury which by turn could lead to renal failure. This could be prevented through rational drug use which would prevent a lot of patients’ suffering. The manuscript was well written and require certain modifications to be performed through the following comments;  

1- Study flow chart was missed. Please add a study flow chart showing the flow of the study including allocation, intervention and analyses.

2-In methodology section the authors did not mention anything concerning the inclusion and exclusion criteria. Inclusion and exclusion criteria as subtitle SHOULD added and mentioned comprehensively.

 3-The medication history and medical history were missed, theses are considered a very vital and important items that affects patients status and might lead to patients’ kidney injury and are considered as a confounders in the present study. Authors had to mention it as a limitation under limitation section.

4-Supplementation, herbal medications and self- medication was missed here which might affect the incidence or kidney injury and thus affects the percentages of reported adverse effects of the study as those with absence of kidney injury might be related to co-administration of theses supplements containing antioxidants which are of great benefits to the patients’ health status. The authors SHOULD add this as a limitation.

5-Page 2, line 92, the author starts paragraph with an acronym “AEs were coded…….”.AE to be written completely as “Adverse events”. Kindly do not start any paragraph with an Acronym as well as do not add acronyms in the figures or table titles if applicable here.

5-Page 7, in Table 4, what did the authors mean by “Systemic diseases and conditions related to the site of administration”?

6-Page 8, lines 177, 178, 179 and 180, the following paragraph to be omitted as it was repeated BY MISTAKE for two times “Figure 1. ROR related to age, sex, and access in AE for renal manifestations, as well as in relation to major classes of concomitant medications.”

7-Page 9 line 182; “Figure 1. ROR related to age, sex, and access in AE for renal manifestations, as well as in relation to major classes of concomitant medications”. Please write ROR as full name and not as acronym.

8-The authors had to explain and rationalize the meaning of using this expression known as “triple whammy”, in other words why Whammy special? and what is its relation with the study?

9-In the discussion section the authors mentioned that most of the triple whammy-related AEs leading to ED access 199 were associated with drug interactions (86.3%), although they did not mention the types of these interactions or classified them according to severity, types, …etc. in a separate table in the study, being 86.3% which is a large percentage, and give some examples of these drug interactions. The authors had to explain why this important information was missed?.

10-The mechanism and pathways by which the ACI, ARBs, Diuretics and NSAID could lead to kidney injury was missed. These mechanisms SHOULD be added in the discussion section to explain why kidney injury might occur.

11-Paragraphs concerning the importance of supplements and or antioxidants which are considered a reno-and cardio-protective agents were missed in the introduction which are considered a very important information to be given to the readers and recommended as a solution to decrease the incidence of kidney injury in patients administering such medication either separately or in as polypharmacy. Please the following paragraphs should be added in the introduction;

   Oral administration of vitamin C/ rutin showed a favorable synergistic effect and might provide a positive future result associated with promise attenuation of inflammation and oxidative stress in end stage renal patients (1), besides, febuxostat improved hyperuricemia and endothelial dysfunction and ameliorate inflammation with no safety concerns (2). Moreover, supplementation with omega-3 in patients with end-stage renal disease showed a highly significant reduction in total cholesterol and a highly significant increase in glutathione peroxidase and superoxide dismutase levels proving the omega-3' beneficial effect on serum lipid profile and oxidative stress.(3) Silymarin with its antioxidant and anti-inflammatory properties was used as an adjuvant therapy for glycemic control, lipid profile and insulin resistance.(4) Additionally, Daflon 500 mg (micronized purified flavonoid fraction of Rutaceae aurantiae) proved to be helpful in reducing glucose level and the risk of cardiovascular disease.(5)

References

(1)    Samia Omar, Radwa Maher El Borolossy, Tamer Elsaid and Nagwa A. Sabri. Evaluation of the combination effect of rutin and vitamin C supplementation on the oxidative stress and inflammation in hemodialysis patients.  Frontiers in Pharmacolology (2022), 13:961590. doi: 10.3389/fphar.2022.961590.

(2)    Alshahawey M., Shahin S.M., Elsaid T.W., Sabri N.A. Effect of febuxostat on the endothelial dysfunction in hemodialysis patients: a randomized, placebo-controlled, double-blinded study.Am J Nephrol 2017;45:452-459. https://doi.org/10.1159/000471893.

(3)    Areej Mohamed Ateya, Nagwa Ali Sabri, Ihab El Hakim, Sara M Shaheen. Journal of Renal Nutrition, Vol. 27 (3), 169-174 (2017). https://doi.org/10.1053/j.jrn.2016.11.005.

12-Concerning hospitalization, the authors have to mention the reasons of hospitalization and how the patient was managed in this case.

Author Response

Reviewer #3:

The manuscript is dealing with a very important topic by considering the adverse effects that might occur through polypharmacy and lead to kidney injury which by turn could lead to renal failure. This could be prevented through rational drug use which would prevent a lot of patients’ suffering. The manuscript was well written and require certain modifications to be performed through the following comments; 

We would like to thank Reviewer 3 for his/her consideration regarding our paper.

1- Study flow chart was missed. Please add a study flow chart showing the flow of the study including allocation, intervention and analyses.

Resolved, restructuring all the paragraphs of the Methods section.

2-In methodology section the authors did not mention anything concerning the inclusion and exclusion criteria. Inclusion and exclusion criteria as subtitle SHOULD added and mentioned comprehensively.

Inclusion and exclusion criteria are reported in the restructured Methods section.

 3-The medication history and medical history were missed, theses are considered a very vital and important items that affects patients status and might lead to patients’ kidney injury and are considered as a confounders in the present study. Authors had to mention it as a limitation under limitation section.

Resolved.

4-Supplementation, herbal medications and self- medication was missed here which might affect the incidence or kidney injury and thus affects the percentages of reported adverse effects of the study as those with absence of kidney injury might be related to co-administration of theses supplements containing antioxidants which are of great benefits to the patients’ health status. The authors SHOULD add this as a limitation.

Resolved.

5-Page 2, line 92, the author starts paragraph with an acronym “AEs were coded…….”.AE to be written completely as “Adverse events”. Kindly do not start any paragraph with an Acronym as well as do not add acronyms in the figures or table titles if applicable here.

Resolved.

5-Page 7, in Table 4, what did the authors mean by “Systemic diseases and conditions related to the site of administration”?

This is a specific SOC (system organ class) term defined by MedDRA classification system, generally used in pharmacovigilance studies for describing adverse events and adverse drug reactions.

6-Page 8, lines 177, 178, 179 and 180, the following paragraph to be omitted as it was repeated BY MISTAKE for two times “Figure 1. ROR related to age, sex, and access in AE for renal manifestations, as well as in relation to major classes of concomitant medications.”

Resolved.

7-Page 9 line 182; “Figure 1. ROR related to age, sex, and access in AE for renal manifestations, as well as in relation to major classes of concomitant medications”. Please write ROR as full name and not as acronym.

Resolved.

We moved Figure 1 in the Supplementary section.

8-The authors had to explain and rationalize the meaning of using this expression known as “triple whammy”, in other words why Whammy special? and what is its relation with the study?

Resolved.

9-In the discussion section the authors mentioned that most of the triple whammy-related AEs leading to ED access were associated with drug interactions (86.3%), although they did not mention the types of these interactions or classified them according to severity, types, …etc. in a separate table in the study, being 86.3% which is a large percentage, and give some examples of these drug interactions. The authors had to explain why this important information was missed?

Resolved.

10-The mechanism and pathways by which the ACE-I, ARBs, Diuretics and NSAID could lead to kidney injury was missed. These mechanisms SHOULD be added in the discussion section to explain why kidney injury might occur.

The mechanism is well described in the Introduction section.

11-Paragraphs concerning the importance of supplements and or antioxidants which are considered a reno-and cardio-protective agents were missed in the introduction which are considered a very important information to be given to the readers and recommended as a solution to decrease the incidence of kidney injury in patients administering such medication either separately or in as polypharmacy. Please the following paragraphs should be added in the introduction;

Oral administration of vitamin C/ rutin showed a favorable synergistic effect and might provide a positive future result associated with promise attenuation of inflammation and oxidative stress in end stage renal patients (1), besides, febuxostat improved hyperuricemia and endothelial dysfunction and ameliorate inflammation with no safety concerns (2). Moreover, supplementation with omega-3 in patients with end-stage renal disease showed a highly significant reduction in total cholesterol and a highly significant increase in glutathione peroxidase and superoxide dismutase levels proving the omega-3' beneficial effect on serum lipid profile and oxidative stress.(3) Silymarin with its antioxidant and anti-inflammatory properties was used as an adjuvant therapy for glycemic control, lipid profile and insulin resistance.(4) Additionally, Daflon 500 mg (micronized purified flavonoid fraction of Rutaceae aurantiae) proved to be helpful in reducing glucose level and the risk of cardiovascular disease.(5)

References

(1)    Samia Omar, Radwa Maher El Borolossy, Tamer Elsaid and Nagwa A. Sabri. Evaluation of the combination effect of rutin and vitamin C supplementation on the oxidative stress and inflammation in hemodialysis patients.  Frontiers in Pharmacolology (2022), 13:961590. doi: 10.3389/fphar.2022.961590.

(2)    Alshahawey M., Shahin S.M., Elsaid T.W., Sabri N.A. Effect of febuxostat on the endothelial dysfunction in hemodialysis patients: a randomized, placebo-controlled, double-blinded study.Am J Nephrol 2017;45:452-459. https://doi.org/10.1159/000471893.

(3)    Areej Mohamed Ateya, Nagwa Ali Sabri, Ihab El Hakim, Sara M Shaheen. Journal of Renal Nutrition, Vol. 27 (3), 169-174 (2017). https://doi.org/10.1053/j.jrn.2016.11.005.

Resolved, adding this new paragraph in the Discussion section.

12-Concerning hospitalization, the authors have to mention the reasons of hospitalization and how the patient was managed in this case.

The events considered to be the cause of the hospitalisation are reported in Table 4. The MEREAFaPS project is an active monitoring of suspected adverse drug reactions as cause of ED visit (the monitoring was performed inside the ED), thus information on patients hospitalised are not available. All the reported patients’ information is collected in ED, no information is retrievable from the hospital wards. Please see our previous publications (https://pubmed.ncbi.nlm.nih.gov/?term=mereafaps&sort=date).

Reviewer 4 Report

In the present manuscript the author study hospitalization related to the combination of ACE inhibitors/or sartans with diuretics and NSAIDs.

The objective of the study is interesting, however there are several important limitations in it.

First the study is based in voluntary reports of pharmacovigilance. It is quite difficult to determine specifically different assertions that the authors make in the manuscript derived from such reports. 

As they show in the results, the patients have diverse comorbidities and are quite polymedicated, therefore is difficult to attribute the problems of these patients solely to the interactions between the drugs objective of the study. Moreover, as it is reported in table 4 the adverse effects are very diverse.

I consider that as the authors are interested in the combination ACE inhibitors/or sartans with diuretics and NSAIDs, they should focus they results to these drugs in a more centerd way.

Minor things:

-       Table 1: Administration routes are oral and systemic, but oral route is a systemic route. Were the authors referring to parenteral rout, when naming it systemic?

Author Response

Reviewer #4:

In the present manuscript the author study hospitalization related to the combination of ACE inhibitors/or sartans with diuretics and NSAIDs.

The objective of the study is interesting, however there are several important limitations in it.

We would like to thank Reviewer 4 for his/her consideration regarding our paper.

First the study is based in voluntary reports of pharmacovigilance. It is quite difficult to determine specifically different assertions that the authors make in the manuscript derived from such reports.

As already reported in the first part of the Methods section, the data analysed in this study come from a national "active" pharmacovigilance project (the MEREAFaPS project). During the study period (active monitoring in EDs), all pharmacovigilance report forms were collected by trained healthcare professional monitors. Readers can find more details about the MEREAFaPS project in our previous publications.

As they show in the results, the patients have diverse comorbidities and are quite polymedicated, therefore is difficult to attribute the problems of these patients solely to the interactions between the drugs objective of the study. Moreover, as it is reported in table 4 the adverse effects are very diverse.

We added this important consideration in the Limitations section.

I consider that as the authors are interested in the combination ACE inhibitors/or sartans with diuretics and NSAIDs, they should focus they results to these drugs in a more centerd way.

As already reported in the Methods section (Exposures), we considered only pharmacovigilance report forms with ACE-I (ATC classes: C09A*/C09B*) and/or ARBs (ATC classes: C09C*/C09D*), diuretics (ATC classes: C03*) and NSAIDs (or Coxibs) (ATC classes: M01A*), focusing the entire analysis on these suspected medications.

Minor things:

- Table 1: Administration routes are oral and systemic, but oral route is a systemic route. Were the authors referring to parenteral rout, when naming it systemic?

Resolved.

Reviewer 5 Report

Thank you for your submission. However, this is a very low-quality study that adds nothing to the current literature. The data are very old and of little interest.

Eighty patients over 11 years? Seven per year - is this really a major clinical issue based on these small numbers?

The major weakness of the study is the lack of any validation of the reported adverse reactions; this is scientifically unacceptable. The manuscript should include a rigorous assessment process for evaluating the potential adverse drug reaction/interaction in each of the 80 reported cases e.g. using a Naranjo score or other causality assessment tool. 

The statistical analysis is also flawed. There is no sample size calculation, but a sample size calculation (e.g. method of Peduzzi) will clearly indicate that the sample size is inadequate for a robust multivariable analysis. Details of the multivariate logistic analysis are also inadequate. How was the issue of collinearity/multicollinearity between variables in the multivariate statistical analyses dealt with (assessed and appropriately managed)? The huge confidence intervals shown in Figure 1 also illustrate the futility of the analysis.

There are too many tables and figures, which (like the statistical analysis) add no value.

The English needs close attention. For example, this sentence is too long and convoluted:

“In this regard, the elderly is at a higher risk of incurring AEs due to drug interactions not only because of the number of drugs taken but also because of the natural aging process that can result in physiological changes such as decreased kidney and liver function, which can affect drug pharmacodynamics and pharmacokinetics [4].”

Please note that polypharmacy is not necessarily a problem in the elderly; inappropriate polypharmacy. For instance, current international guidelines recommend 4 drugs as base therapy for congestive heart failure; similar recommendations apply to type 2 diabetes and ACS. All these individuals will invariably (and should) have polypharmacy. A lack of polypharmacy would indicate sub-therapeutic management.

Author Response

Reviewer #5:

Thank you for your submission. However, this is a very low-quality study that adds nothing to the current literature. The data are very old and of little interest.

We would like to thank Reviewer 5 for his/her consideration regarding our paper.

Eighty patients over 11 years? Seven per year - is this really a major clinical issue based on these small numbers?

We added this important consideration in the first part of the Discussion section.

The major weakness of the study is the lack of any validation of the reported adverse reactions; this is scientifically unacceptable. The manuscript should include a rigorous assessment process for evaluating the potential adverse drug reaction/interaction in each of the 80 reported cases e.g. using a Naranjo score or other causality assessment tool.

All details, including the methodology applied for the construction of the MEREAFaPS database, can be found in our previous publications [in particular, Lombardi N et al., 2020]. Moreover, all pharmacovigilance report forms analysed here were all validated in terms of causality assessment using specific algorithms and/or scales (i.e., Naranjo scale) as requested by the Italian Medicines Agency and by the Italian Pharmacovigilance System legislation. As already reported, a multidisciplinary team composed by experts in clinical pharmacology, toxicology, and pharmcoepidemiology, performed a clinical evaluation of cases included in the MEREAFaPS database, in order to maintain a high-quality standard of the data.

We added this information in the first part of the Methods section (Study design and setting).

The statistical analysis is also flawed. There is no sample size calculation, but a sample size calculation (e.g. method of Peduzzi) will clearly indicate that the sample size is inadequate for a robust multivariable analysis. Details of the multivariate logistic analysis are also inadequate. How was the issue of collinearity/multicollinearity between variables in the multivariate statistical analyses dealt with (assessed and appropriately managed)? The huge confidence intervals shown in Figure 1 also illustrate the futility of the analysis.

Considering the small sample of cases (I.e., a small number of pharmacovigilance report forms reporting all drugs of interest) included in the multivariate logistic regression, the shift of Figure 1 in the Supplementary materials appears to be more appropriate.

There are too many tables and figures, which (like the statistical analysis) add no value.

Considering the small sample of patients analysed, we think that the number of tables is adequate for the description of their main clinical and pharmacological characteristics.

We moved Figure 1 in the Supplementary section.

The English needs close attention. For example, this sentence is too long and convoluted:

“In this regard, the elderly is at a higher risk of incurring AEs due to drug interactions not only because of the number of drugs taken but also because of the natural aging process that can result in physiological changes such as decreased kidney and liver function, which can affect drug pharmacodynamics and pharmacokinetics [4].”

Resolved.

Please note that polypharmacy is not necessarily a problem in the elderly; inappropriate polypharmacy. For instance, current international guidelines recommend 4 drugs as base therapy for congestive heart failure; similar recommendations apply to type 2 diabetes and ACS. All these individuals will invariably (and should) have polypharmacy. A lack of polypharmacy would indicate sub-therapeutic management.

We added this important consideration in the Conclusions section.

Round 2

Reviewer 4 Report

The authors have made some clarifications and improvements in the manuscript. 

Author Response

We would like to thank Reviewer 4 for his/her final considerations regarding our paper.

Reviewer 5 Report

Thank you for the revision. The statistical analysis is still flawed. Conducting a multivariable analysis with 24 cases of the outcome (hospitalisation) is worthless. And how was the issue of collinearity/multicollinearity between variables in the multivariate statistical analysis dealt with (assessed and appropriately managed)? Please delete the multivariate statistical analysis and Supp F1. They add nothing and simply highlight poor scientific understanding.

Author Response

We thank the Reviewer for this comment and for the opportunity to discuss this issue. We agree with the Reviewer in recognizing that this multivariate model is limited by the small number of cases populating it. Nevertheless, we believe that keeping it (at least as a Supplementary material) might be useful for the reader, to provide a trend on the association between the considered variables and the outcome. However, as we are worthy of this limitation, we addressed this aspect in the discussion, as follows:

Limitations section

“As a final point, the results of the multivariate analysis are limited by the small number of events reported in this population”.

Regarding the presence of multicollinearity in the regression model, we excluded it by calculating the Variance Inflation factor (VIF). We better specified this methodological aspect in the Methods, as follows:

Methods section

“The presence of multicollinearity in the multivariate regression model was excluded by calculating the variance inflation factor (VIF)”.